# The automated Galaxy-SynBioCAD pipeline for synthetic biology design and engineering

Joan Hérisson[1,9], Thomas Duigou [2,9], Melchior du Lac[2,3], Kenza Bazi-Kabbaj[2], Mahnaz Sabeti Azad[2], Gizem Buldum[4], Olivier Telle[2], Yorgo El Moubayed[2], Pablo Carbonell [5,6], Neil Swainston[5,7], Valentin Zulkower[8], Manish Kushwaha [2], Geoff S. Baldwin [4] & Jean-Loup Faulon[1,2,5] ✉

Here we introduce the Galaxy-SynBioCAD portal, a toolshed for synthetic biology, metabolic engineering, and industrial biotechnology. The tools and workflows currently shared on the portal enables one to build libraries of strains producing desired chemical targets covering an end-to-end metabolic pathway design and engineering process from the selection of strains and targets, the design of DNA parts to be assembled, to the generation of scripts driving liquid handlers for plasmid assembly and strain transformations. Standard formats like SBML and SBOL are used throughout to enforce the compatibility of the tools. In a study carried out at four different sites, we illustrate the link between pathway design and engineering with the building of a library of *E. coli* lycopene-producing strains. We also benchmark our workflows on literature and expert validated pathways. Overall, we find an 83% success rate in retrieving the validated pathways among the top 10 pathways generated by the workflows.

Computation has become an essential tool in life science research. Synthetic biology, metabolic engineering and industrial biotechnology make no exception to that trend. As part of this endeavor, significant attention is being paid to the development of workflows adhering to design principles from engineering such as standardization and abstraction of modular parts, as well as the decoupling of design from fabrication.

Following the electronic design automation (EDA) concept, there are many design automation tools for genetic circuits, these are extensively reviewed in Appleton et al.[1]. As an example, Cello[2] applies the EDA approach to genetic circuits. Cello comprises several steps, which are connected and therefore need to use standardized input/output formats. Among those formats are Verilog to represent a logic

function and Eugene[3] to encode a set of parts and constraints between the parts. Although Cello achieved the compilation and standardization of several pieces of software for genetic design, in general, this is not true for most freely available synthetic biology and metabolic engineering design tools, where the fragmentation remains a significant barrier to adoption. Nonetheless, two main standards have emerged in the past two decades. The first, SBML[4] is a biological modeling standard that has been developed by the systems biology community to encode strains and pathways. The second, SBOL[5], is a data exchange standard specific to synthetic biology. SBOL has been developed to document genetic components (DNA, RNA, protein, etc.) and their interactions for the purpose of biodesign engineering. SBOL can now encode complex genetic circuits, metabolic pathways,

[1]Genomics Metabolics, Genoscope, François Jacob Institute, CEA, CNRS, Univ Evry, Université Paris-Saclay, 91057 Evry, France. [2]Université Paris-Saclay, INRAE, AgroParisTech, Micalis Institute, 78352 Jouy-en-Josas, France. [3]Amyris Inc, Emeryville, CA 94608-2405, US. [4]Imperial College Centre for Synthetic Biology, Department of Life Sciences, Imperial College, London SW7 2AZ, UK. [5]Manchester Institute of Biotechnology, SYNBIOCHEM center, School of Chemistry, The University of Manchester, Manchester M1 7DN, UK. [6]Institute of Industrial Control Systems and Computing (ai2), Universitat Politècnica de València, 46022 Valencia, Spain. [7]Institute of Systems, Molecular and Integrative Biology University of Liverpool, Liverpool L69 7ZB, UK. [8]Edinburgh Genome Foundry, SynthSys, School of Biological Sciences, University of Edinburgh, EH93BF Edinburgh, UK. [9]These authors contributed equally: Joan Hérisson, Thomas Duigou. ✉e-mail: Jean-Loup.Faulon@inrae.fr

vectors, and plasmids. Complying with SBML and SBOL standards a suite of in silico genetic circuit design tools was recently proposed[6].

As for genetic circuits, there are plenty of software tools to assist the biosynthetic pathway design process[7]. Briefly, from a given target compound and a given chassis strain, the first step consists of finding metabolic reactions that link the target compound to the native metabolites of the host strain. This step is carried out by retrosynthesis software tools[8–13] and, should one wish to search for novel pathways or find pathways that produce unnatural target compounds, requires the use of reaction rules[14]. The result of retrosynthesis is a metabolic map and there is a need in a second step to enumerate the pathways linking the chassis metabolites to the target. There are many solutions for pathway enumeration and search[15], which are sometimes integrated into the retrosynthesis software itself. The third step is to find the most promising enzyme sequences catalyzing the metabolic reactions. This can be achieved either through similarity search to enzyme annotated metabolic reactions[16–18], or machine learning trained on metabolic databases[19,20]. Once the pathways have been annotated with enzyme sequences, they can be ranked in a fourth step. The ranking criteria are diverse, they can be among others based on thermodynamics[21], predicted yield of the target[22], target rate of production through flux balance analysis[9,11,21], chassis cytotoxicity of the target and intermediates[21], along with simpler criteria like pathway length.

In addition to the enzyme identities, there are multiple layout solutions and settings to engineer the top-ranked pathways. Indeed, the individual genes coding for the enzyme can be placed under different promoters, in a different order within an operon, with different RBS strength (if the chassis is a bacteria), and on different plasmids with different origins of replication if the engineering is performed on a plasmid. The fifth step deals with this issue by making use of tools such as the RBS calculator[23] to compute RBS sequences for different strengths, and design of experiments (DoE)[24,25] to sample the space of possible constructs, which can be very large. The result of that step is a library of layouts representing either the same or different pathways. At this stage, one can either synthesize the whole layout DNA or, as it is most commonly done, synthesize individual DNA parts and use combinatorial DNA assembly methods to include variations of the control elements, such as promoters and RBS sequences. Several computational tools can be used to perform this *sixth* and last step of assembly design before constructing the pathways, these tools compute parts to be synthesized depending on the chosen assembly protocol. Computation tools to help the build tasks are sparser than for design. One can cite here Aquarium[26], which provides instructions to a person or a robot to perform the assembly tasks along with Antha[27], BioBlocks[28], and DNA-BOT[29]. Engineered pathways are generally evaluated using HPLC or mass spectrometry analyses. Here too, computational tools can help in particular the workflows produced by OpenMS[30] or Worlflow4Metabolomics[31].

Considering the above, we are clearly at a stage where the pathway engineering process is not that far from being fully driven by computer software products. However, there are several hurdles that prevent this from happening even for tools covering pathway design only. First, the tools are not easily findable, they are stored in different places and the keywords to search online are not obvious. Secondly, some of the tools are difficult to access, some requiring registration, purchase, or access fees. Thirdly, almost none of the tools are interoperable and cannot be chained one after another to ensure that computational experiments are communicated well, and hence reproducible. Lastly, and perhaps most problematic for wider acceptance, the tools can be difficult to comprehend, requiring a level of expertize that limits their use by a large community.

Scientific workflows help to address these issues by providing an open, web-based platform for performing findable and accessible data analyses linked to experimental protocols for all scientists irrespectively of their informatics expertize, along with interoperable and reproducible computations regardless of the particular platform that is being used[32]. Indeed, without programming skills, scientists that need to use computational approaches are impeded by difficulties ranging from tool installation to determining which parameter values to use, to efficiently combining and interfacing multiple tools together in an analysis chain. Scientific workflows provide solutions where data is combined and processed into a configurable, structured set of steps. Existing systems often provide graphical user interfaces to combine different technologies along with efficient methods for using them, and thus increase the efficiency of the scientists using them. In addition, workflow systems generally provide a platform for developers seeking a wider audience and broad integration of their tools, and can thus drive forward further developments in a specific field of research. Among existing workflow platforms, Galaxy is a system originally developed for genome analysis[33] which now includes more than 8500 tools that can be found in the public ToolShed[34].

Here, we introduce the Galaxy-SynBioCAD portal[35], a Galaxy set of tools for synthetic biology, metabolic engineering and industrial biotechnology. It allows one to easily create workflows from the incorporated toolset or use already developed shared workflows. The portal is a growing community effort where developers can add new tools and users can evaluate the tools performing design for their specific projects. The tools and workflows currently shared on the Galaxy-SynBioCAD portal cover an end-to-end metabolic pathway design and engineering process from the selection of strain and target to automated DNA parts assembly and strain transformation.

## Results

### Tools selection criteria

To develop an integrated ecosystem, we selected software applications from among the computational tools mentioned above. Several criteria were used for this selection: the tools needed to (i) be relevant for pathway design and engineering, (ii) be published, (iii) be open-source (MIT, GNU GPL, or related licenses), (iv) be well documented and deposited in GitHub, (v) make use of standard input/output, and (vi) exist as a standalone command-line tool. Within a workflow, each tool connected to one or more tools must share common file format for data exchange, i.e., each output file of a tool has to be compatible with the input file format of downstream tools in the workflow. The file format relies on the nature of the data (e.g., metabolic model, metabolic pathway, and construct design) and the implementation choice made for each tool. Among the standard formats used, some are rather generic (CSV, TSV, JSON) while others are more specific to a scientific field (e.g., SBOL, SBML).

### Pathway design and engineering tools and workflows

The selected tools are further described in the 'Supplementary_Text' file (cf. Galaxy-SynBioCAD Tools). These tools can be divided in three categories: (i) those aimed at finding pathways to synthesize heterologous compounds in chassis organisms (RetroRules, RetroPath2.0, RP2Paths, and rpCompletion), (ii) those aimed at evaluating and ranking pathways (rpThermo, rpFBA, rpReport, rpViz, and rpScore) and (iii) those related to genetic design and engineering (Selenzyme, SbmlToSbol, PartsGenie, OptDOE, DNA Weaver, LCR Genie, rpBASICDesign, and DNA-Bot). Following FAIR principles[32], all selected tools are open-source with code available on GitHub and installable through the Conda package manager[36] (cf. Tools design and integration process in the 'Supplementary_Text' file). Therefore, any user can install the tools needed on their own computer and run these as standalone programs or chain them together to process more complex calculations.

To go further in chaining tools, three types of Galaxy workflows are available on the Galaxy-SynBioCAD portal.
1. A Retrosynthesis workflow to enumerate the pathways enabling the synthesis of a given target chemical in a host chassis organism

(cf. Retrosynthesis from target to chassis in Methods section and Retrosynthesis workflow in 'Supplemetary_Text' file).

2. A Pathway analysis workflow to score and rank the pathways produced through Retrosynthesis step based on multiple criteria (cf. Pathway analysis workflow in 'Supplemetary_Text' file). Some criteria necessitated the development of specific methods for pathways thermodynamics (cf. Thermodynamics in Methods section) and theoretical product flux calculation (cf. Flux Balance Analysis with Fraction of Reaction in Methods section). Pathway scoring is performed via Machine Learning using a training set of pathways extracted from literature and pathways validated by pathway engineering experts (cf. subsection Benchmarking workflows with literature data and expert validation trial).

3. Two Genetic design and engineering workflows that produce assembly plans for plasmids encoding the pathways generated by the Retrosynthesis or Pathway analysis workflows (cf. Genetic design and engineering workflows in 'Supplemetary_Text' file). The first workflow generates plans for Golden Gate[37], Gibson[38], and Ligation Chain Reaction (LCR)[39] assembly methods. It also includes Design of Experiment (OptDoE tool) for combinatorial experimental design. The second workflow (BASIC assembly) generates plans for the Biopart Assembly Standard for Idempotent Cloning (BASIC) technique[40]. This workflow provides a direct link between machine-enabled design and automated assembly. It takes as input a pathway and generates a script to operate an Opentrons liquid handler robot performing assembly and chassis transformation.

In the reminder of the paper, a metabolic pathway is a succession of chemical reactions transforming reactants into products, while a construct is the assembly of genetic parts that encodes a metabolic pathway. Several constructs with different enzyme and regulatory sequences can encode for the same pathway. The Retrosynthesis and Pathways analysis workflows generate annotated SBML files describing pathways (cf. Pathway annotation in Methods section), which are taken as input to the Genetic design and engineering workflows to produce constructs, i.e., plasmids encoded in SBOL format along with assembly plans (in CSV files) and liquid handler instructions (python scripts directly executable by Opentrons).

## Benchmarking workflows for Lycopene production

The *Retrosynthesis* workflow was run at the Genoscope laboratory (Paris region, France) for the production of lycopene in *E. coli*. We used iML1515[41] as a model for *E. coli*. The retrosynthetic map was composed of 12 unique compounds and 7 unique reactions, resulting in 3 "master" pathways (cf. Retrosynthesis from target to chassis in Methods section). Only the 10 best pathways were kept per master pathway, after reaction completion with cofactors and removal of duplicates only 9 pathways remained. Additional details are provided in the Supplementary file 'Dataset 1'.

The Pathway Analysis workflow was run at the University Polytechnic of Valencia, Spain. The workflow took as inputs the list of nine pathways generated by the Retrosynthesis workflow. Results are shown in Fig. S6 in the 'Supplementary_Text' and 'Dataset 1' files. The top-ranked pathway was composed of 3 reactions with EC numbers listed from chassis metabolites to target: 2.5.1.29, 2.5.1.96 and 1.3.99.31.

The Genetic design and engineering workflow for BASIC assembly was run at two different locations: Paris (Micalis Institute) and London (Imperial College). In both cases, as design input we used the top lycopene ranked pathway predicted by the Pathway analysis. Constraining the enzyme search within the organism *Pantoea ananas*, enzymes CrtE (UniProt ID: P21684), CrtB (P21683), and CrtI (P21685) were predicted by the Selenzyme tool for the three-reaction pathway (Fig. S8 in the 'Supplementary_Text' file and 'Dataset 1' Supplementary file). A total of 88 construct designs were automatically generated by

the genetic design and engineering BasicDesign tool. The designs were coded in a CSV file that was fed to DNA-Bot, which was executed in Paris and London with different labware identifiers and associated parameters (cf. Genetic design and engineering workflow execution in Methods). In both laboratories, DNA-Bot generated four executable python scripts (clip reactions, purification, assembly and strain transformation) that were run on Opentrons robots. Additional information can be found in the Supplementary file 'Dataset 1'.

Following the executable scripts produced by the Genetic design and engineering workflow the three genes of the pathway (*crtE*, *crtB* and *crtI*) were assembled in varying gene order in an operon, together with six different RBS-linkers (Fig. 1.a). Each of these linkers held a ribosome-binding site of high or low strength for each of the three genes (cf. Supplementary file 'Dataset 1'); the remaining untranslated region upstream of the RBS provides the overlap sequence that drives the assembly. For consistent RBS context the three untranslated sequences of the linkers were always upstream of the same gene in the assembly. The pathway operon was expressed using one of two different promoters (medium strength PJ23105 and low strength PJ23116).

In both laboratories (Paris and London) the scripts were used for the 88 constructs and to spot 10 μL of the transformed cells (*E. coli* DH5-alpha) on a rectangular LB-agar plate (cf. Lycopene production in Methods section). Of these, only 30 (22 red + 8 white) and 33 (21 red + 12 white) constructs gave transformant colonies at Paris and London respectively (Fig. 1b). However, only 12 (11 red + 1 white) of these transformants were common across the two laboratories, suggesting that 10 μL may be too low a volume to spot for these transformations. To test if more transformants can be obtained by increasing this volume, we manually plated 100 μL of the transformed cells in Paris and repeated the spotting step in London using 40 μL on a 12-well plate (cf. Lycopene production in Methods). Of the 88 constructs, this time transformants were obtained for 51 (41 red + 10 white) and 63 (49 red + 14 white) constructs at Paris and Imperial, respectively, including 36 (33 red + 3 white) constructs in common.

An analysis of the number of successful transformants obtained in the two laboratories for the different combinations of promoter, RBS, and gene order indicates a preference for the weaker J23116 promoter (Fig. 1c). Overexpression of the three pathway genes from a strong promoter may be too toxic for the cell, resulting in overall reduced fitness and consequently fewer successful transformants. Four transformant colonies with visibly different levels of red color (Fig. 1e) were used for acetone extraction of lycopene at Micalis (cf. Genetic design and engineering workflow execution in Methods), and similarly eight colonies were used for lycopene extraction at Imperial. The highest lycopene production was obtained for construct G6 (4.389 mg/gDCW), a yield comparable to those from *E. coli* in similar conditions (5.69 mg/gDCW from *E. coli* DH5-alpha in 2xYT[42] and 6.52 mg/gDCW from *E. coli* ATCC 8739 in LB[43]). In both the weak (J23116) and the high (J23105) promoter groups, low lycopene production was observed from constructs with more than one high-strength RBS (Fig. 1d). When comparing the constructs with the same gene order, for example *crtE-crtI-crtB* (H10, B9), *crtB-crtI-crtE* (B11, D2), or *crtI-crtE-crtB* (C9, D8), constructs with more low-strength RBSes exhibited higher lycopene production. There was also an apparent preference for the *crtI-crtE-crtB* (G6, C9, D8) among the highest producing constructs. Taken together, these data indicate that maximizing the expression of pathway genes can increase cellular burden, resulting in lower pathway productivity.

## Benchmarking workflows with literature data and expert validation trial

Criteria computed by the Pathway analysis workflow like target product flux, thermodynamic feasibility, pathway length, and enzyme availability score inform the user as to the best potential candidate

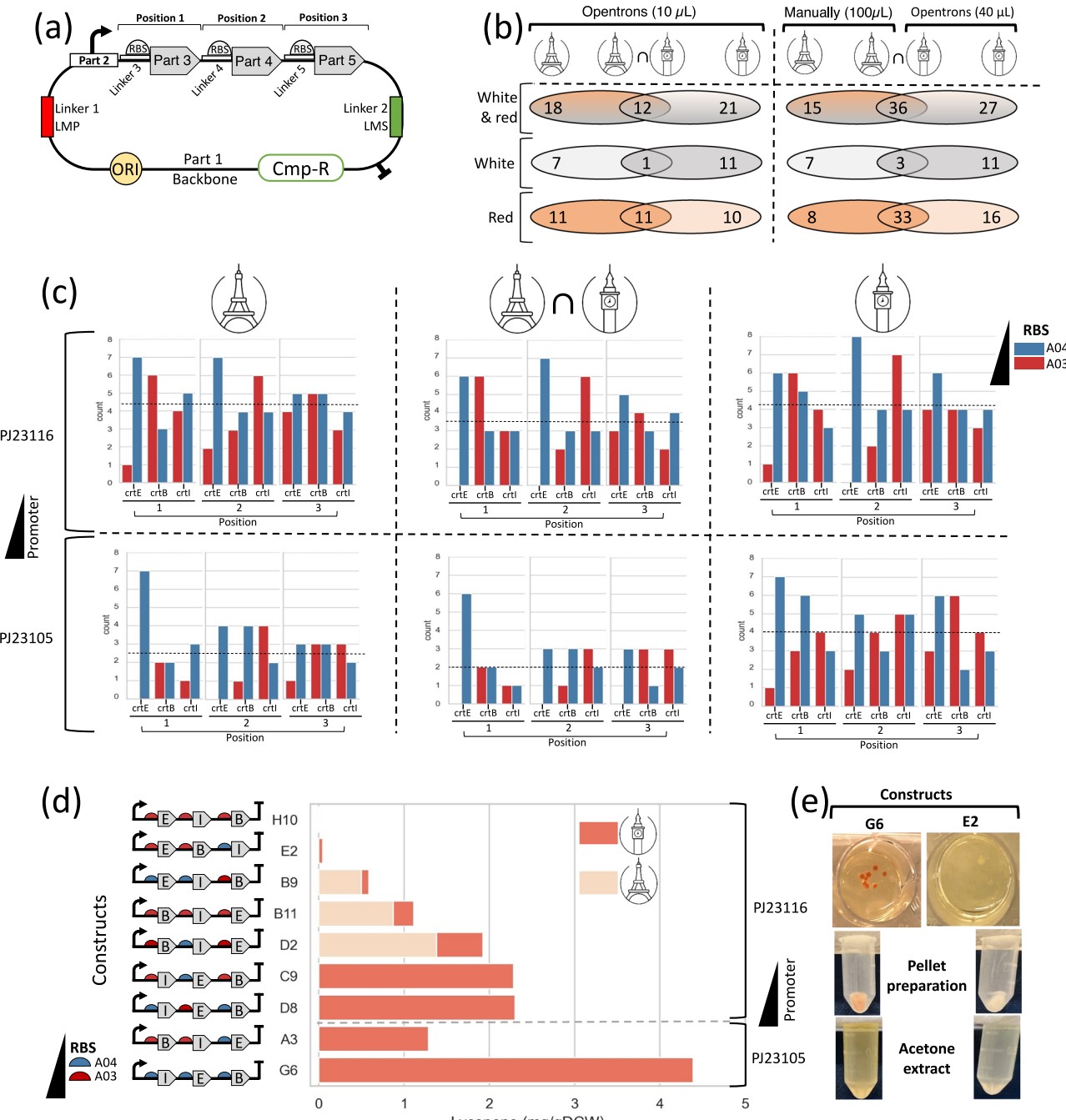

**Fig. 1 | Automated construction of 88 distinct plasmids coding for lycopene pathway operons containing genes in different orders, with varying promoters and RBSes. a** Plasmids coding for lycopene genes were assembled using the BASIC method (with DNA linkers). Genes in the lycopene pathway, *crtE*, *crtB*, and *crtI* (parts 3, 4, and 5), were assembled in an operon with UTR-RBS linkers containing different RBSes. The promoter (part 2) and the 3-gene operon were assembled into a backbone with p15A origin of replication (ORI) and chloramphenicol resistance gene (Cmp-R). The assembled parts were flanked by methylated linkers that recapitulate BASIC prefix and suffix (LMP and LMS). **b** Number of constructs with successful transformant colonies and their color are reported from Micalis Institute, Paris, and Imperial College, London. Number of constructs common to both laboratories are in the intersection. Data are from spotting 10 μL of transformation reactions by Opentrons (left) and from spotting 100 μL manually or 40 μL by Opentrons (right) on LB plates. **c** Count-plots show the number of constructs with successful colonies, grouped by position and gene (details in Supplementary file 'Dataset 1'). Results are from Paris (left), from London (right), and in common (middle). Constructs have a weaker promoter (top) or stronger promoter (bottom). The RBSes are differentiated by colors. The genes' positions in the operon are indicated on the x-axis. Means of the number of constructs for each promoter are shown by dashed lines. **d** Lycopene measurement (mg of lycopene per gDCW) from different constructs from both laboratories. Types of RBSes and promoters, and gene orders are indicated. E: *crtE*, B: *crtB*, I: *crtI*. Promoters and terminators are shown at the extremities. **e** Examples of red and white colonies (top), pellet preparation (middle) and acetone extracted lycopene (bottom). Source data are provided in the 'Source Data' file.

pathway to produce a compound of interest. These criteria can be combined in a global score value. To that end, we developed a machine learning scoring tool (cf. Machine Learning Global Scoring in Methods) taking training data from literature and a validation trial conducted by metabolic engineering experts (cf. Acknowledgement section for the list of experts enrolled). The process is summarized in Fig. 2, the literature benchmarking and expert trial results are provided in the following subsections.

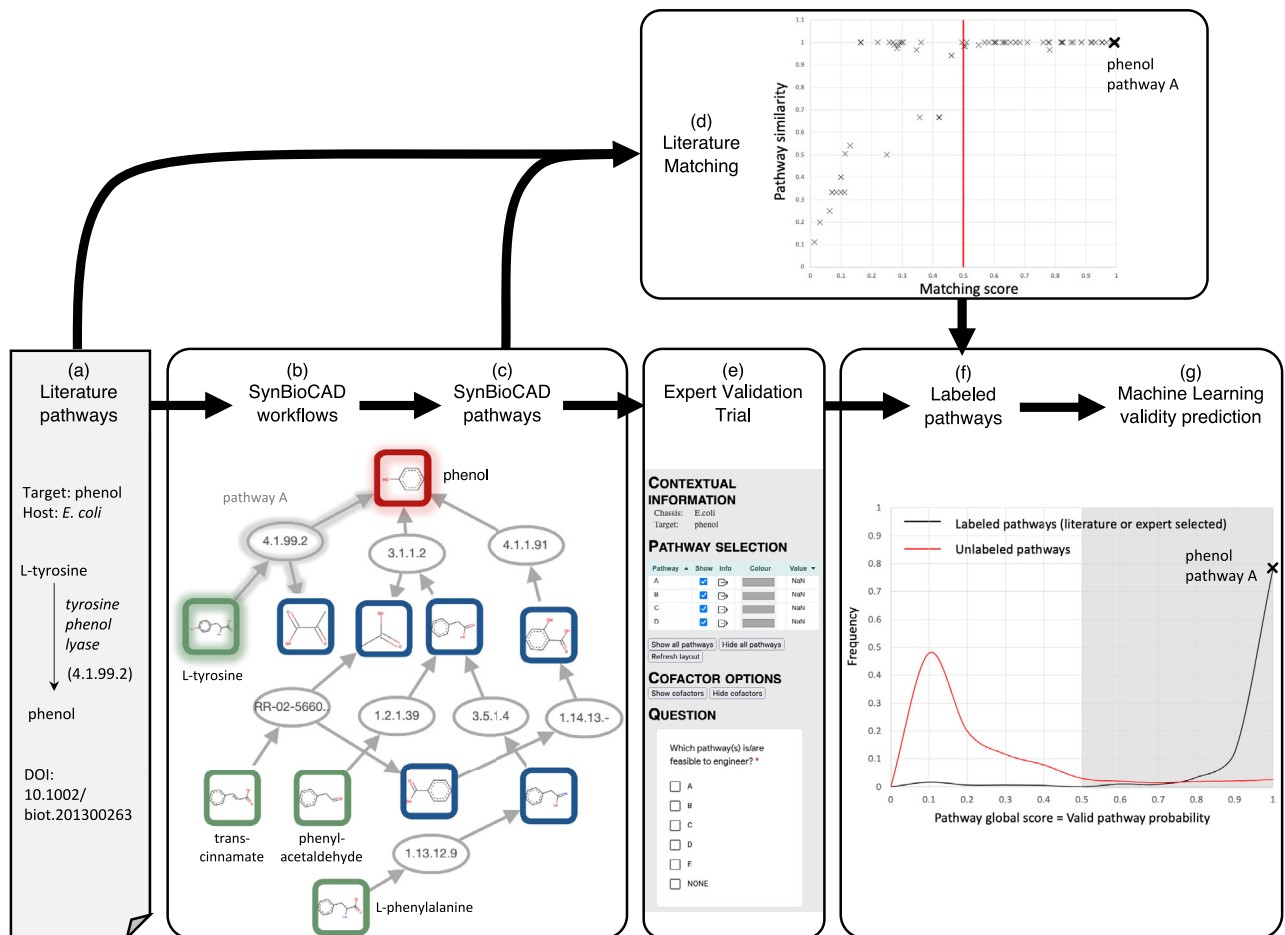

**Fig. 2 | Scoring Galaxy-SynBioCAD predicted pathways with literature pathways and expert validation data. a** Pathways for different targets and different hosts are extracted from literature (cf. Literature data benchmarking subsection), this is illustrated here for production of phenol in *E. coli*. **b** Galaxy-SynBioCAD workflows are run on the literature targets and hosts. **c** A collection of Galaxy-SynBioCAD generated pathways is compiled. Pathway 'A' producing phenol in *E. coli* from tyrosine is highlighted. **d** The Galaxy-SynBioCAD generated pathways are compared with the literature pathways using a matching algorithm (cf. 'Supplementary_Text' file). The plot shows for each literature pathway the best matching pathways among all Galaxy-SynBioCAD generated pathways. Pathways having a matching score above 0.5 are identical (similarity of 1) to literature pathways as far as main substrate and products are concerned. The raw data can be found in Supplementary file 'Dataset 2', tab 'literature_matching_score'. **e** Galaxy-SynBioCAD

generated pathways are evaluated by metabolic engineer experts whose task is to select in batches of 5 generated pathways which ones are valid (cf. Expert validation trial benchmarking subsection). **f** Valid pathways according to experts and pathways matching literature are added to a training set of labeled pathways. **g** The set of labeled pathways is used to train a classifier printing out a machine learning score to assess if a given pathway is valid or not (cf. Machine Learning Global Scoring in Methods section). The figure plots the results obtained for all pathways generated by Galaxy-SynBioCAD. The raw data, including the training set, can be found in the Supplementary file 'Dataset 3'. Using a machine learning global score threshold of 0.5, the accuracy retrieving literature of expert labeled pathways is 0.91 with a false positive rate of 0.10 in 4-fold cross validation (cf. Supplementary file 'Dataset 3', tab 'Pathway_PredictedScore'). Source data are provided in the 'Source Data' file.

**Literature data benchmarking.** A list of 77 pathways corresponding to 60 expressed compounds in engineered organisms (*E. coli, S. cerevisiae*, and *P. putida*) was collected from the literature (cf. Supplementary file 'Dataset 2'). For each of the 77 collected pathways and each heterologous pathway reaction, we compiled the EC number of the reaction along with the corresponding substrates and products. Each target compound within that list was used to run the Retrosynthesis and Pathways analysis workflows to generate a collection of 5874 predicted pathways that produced the same target molecule in the same host organism as those reported in the literature. Following that, the predicted collection of pathways were compared with their corresponding literature pathways using a matching algorithm described in the 'Supplementary_Text' file. Figure 2d shows for each literature pathway, the predicted pathway with the highest matching score (raw data is in Supplementary file 'Dataset 2'). Any pathway generated by Galaxy-SynBioCAD is labeled 'literature pathway' if its score is above 0.5 and that pathway is added to the training set of a machine learning model predicting global score (Fig. 2f, g).

**Expert validation trial benchmarking.** Pathways generated by Galaxy-SynBioCAD should not be discarded even when they do not appear in the literature, for the obvious reason that not all pathways have been engineered for the 60 targets of our literature benchmarking. To palliate this shortcoming, we generated a set of 7919 predicted pathways for 80 (target, chassis) pairs using the Retrosynthesis and Pathway analysis workflows. The set included the 5874 pathways generated for our literature benchmarking along with 2045 additional pathways corresponding to 20 additional (target, chassis) pairs taken from the LASER database[44], which includes some pathways from *B. subtilis*. We next spliced the set in batches of 5 pathways synthesizing the same target in the same chassis. The predicted pathways best matching the literature pathways (when known) were included and the 4 remaining pathways were drawn randomly. We next recruited 40 experts in the metabolic engineering community (see Acknowledgement section) and asked them to select valid pathways in the list they received. To help the selection process, the experts received a clickable map of the 5 pathways (Fig. 2e) where they could

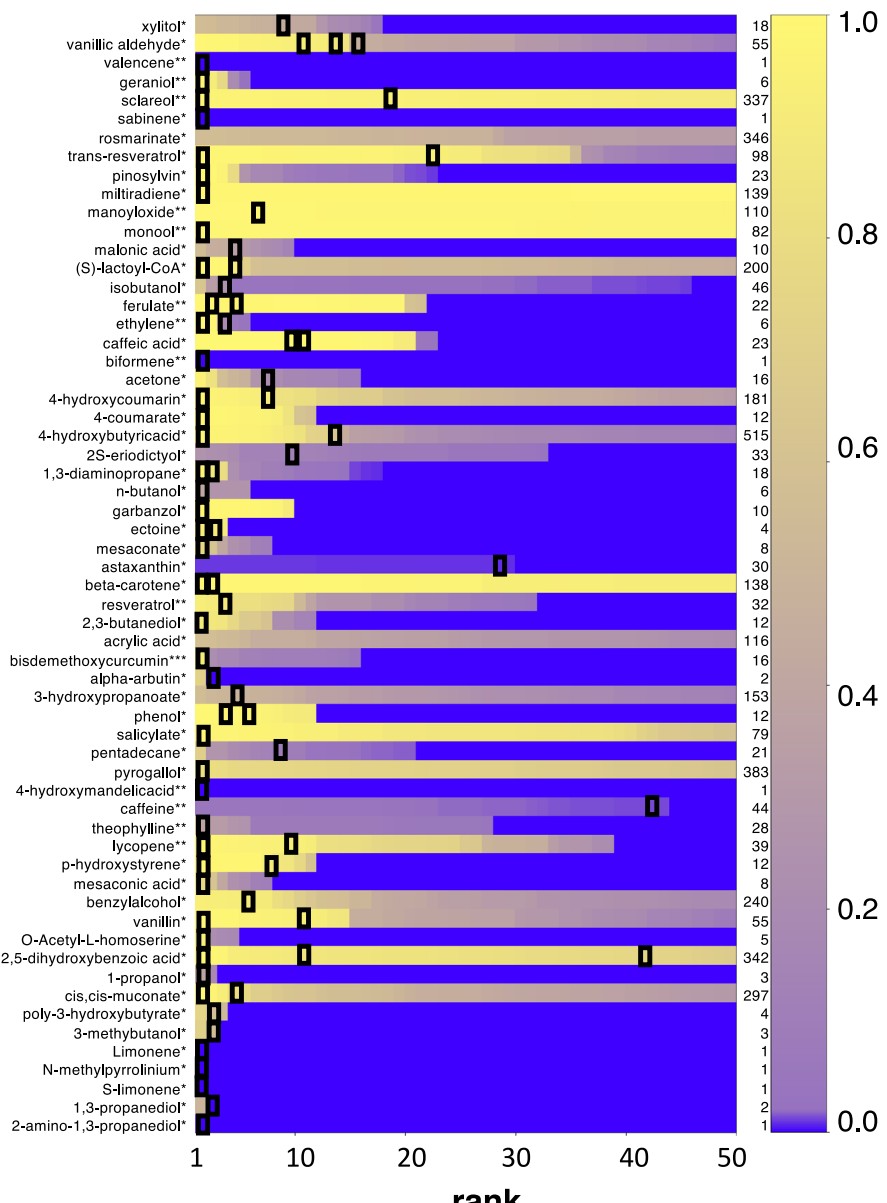

**Fig. 3 | Ranking predicted pathways with machine learning global score.** The color code on the right side shows the machine learning global score (from 1 top to 0). The black boxes show the location of the literature or expert selected pathways for a set 60 literature target engineered in *E. coli* (*), *S. cerevisiae* (**) or *P. putida* (***). If a row does not contain a black box, then the literature or expert selected pathway is not found within the first 50 scored pathways. The numbers listed on the right side are the total numbers of pathways generated by the Galaxy-SynBioCAD workflows. The data used to generate the figure can be found in Supplementary file 'Dataset 3', tab 'Lit_Pathway_Rank_ML'. Source data are provided in the 'Source Data' file.

collect information on compounds and reactions, reaction and pathway thermodynamics, a ranked list of enzymes catalyzing each reaction, and reaction and pathway production fluxes. An example of such a map can be found on the Galaxy-SynBioCAD portal. The results were recorded and merged with the literature benchmarking results using an OR function for identical pathways. At the end of this process, among the 7919 pathways, 754 were labeled positive either because their matching score with a literature pathway was above 0.5 or because they were selected as feasible to engineer by the experts (cf. Supplementary file 'Dataset 3').

Using the literature and expert validated pathways, we developed a machine learning model whose purpose was to evaluate if any given pathway is a valid one or not. To that end, we used a classifier (cf. Machine Learning Global Scoring in Methods section), which returned a global score for each queried pathway. The global score distribution for the dataset of 7919 pathways is given in Fig. 2g (raw data are in

Supplementary file 'Dataset 3'). The classifier exhibited excellent performances with an average cross-validation accuracy of 0.91.

The machine learning global scoring process was used to rank the top 50 Galaxy-SynBioCAD pathways generated for 60 target molecules taken from our literature pathway training set. More precisely, a global score was calculated for all SynBioCAD generated pathways using machine learning. Concomitantly, literature or expert selected pathways were identified using the matching algorithm described in the 'Supplementary_Text' file. Results are shown in Fig. 3 where each row is a ranked list of collections of SynBioCAD generated pathways for a given target molecule in a given chassis. Literature or expert selected pathways are flagged with a black square and pathways are ranked according to the global score aforementioned. Overall, we find that in 83% cases, the literature or expert selected pathways have a score in the top 10 scores of all the pathways generated for the same target and chassis. The number rises to 94% in retrieving literature or expert

validated pathways among the top 50 machine learning scored pathways.

To assess the advantage of making use of a machine learning scoring schema, we computed a direct score based on (i) the inverse of the pathway length (favoring shorter pathway), (ii) the opposite normalized pathway free energy (favoring high negative values), (iii) the normalized averaged enzyme availability score, and (iv) the normalized FBA calculated product flux value. All these values were calculated using the Pathway analysis workflow (addition information are found in Table 3 in Methods section). A direct score calculation, which does not make use of machine learning, can simply be obtained summing the above four parameters. Results are presented in the Supplementary file 'Dataset 3' (tab 'Lit_Pathway_Rank_noML') where 58% of the identified literature/expert pathways are found within the top 10 scored pathways generated by Galaxy-SynBioCAD, this number is lower than the one obtained when using machine learning (83%).

## Discussion

We have presented several Galaxy workflows to design and engineer pathways in host organisms. These workflows have been built using 25 different computational tools. Chaining the tools together to form workflows was made possible only because the input and output of each tool were standardized. As far as standardization is concerned, we chose community adopted standards like InChI and SMARTS for compounds and reactions, SBML for pathways and strains, and SBOL for genetic constructs.

We illustrated our workflows by designing and engineering a library of 88 pathway variants designed to produce lycopene in *E. coli* DH5-alpha on Opentrons liquid handlers. The workflows were executed at four different locations demonstrating the ability of the Galaxy-SynBioCAD portal to run workflows (including robot drivers with different labwares) at different sites, and consequently the possibility of completing multi-partners design and engineering projects.

There are many standard protocols for biological engineering but as argued elsewhere[45], written protocols without practical guidance can lead to problems, as protocols often contain ambiguities or rely on tacit knowledge. Here the possibility of running several times the same workflow that incorporates automated experiments provides a systematic way to quantify reproducibility (cf. Fig. 1).

To assess the validity of our generated pathways, we used a double-blind testing strategy performed by a pool of participants. In that strategy, used previously to evaluate synthesis planning[46], neither the participants nor the conductors are aware of the origin of the pathways, and the participants are asked to flag pathways they deemed valid without having explicit information on pathways found in literature. We applied this approach to develop a machine learning based scoring function reaching high predictability when ranking pathways.

The Galaxy-SynBioCAD portal presented in this paper, proposes a set of synthetic biology and metabolic engineering computational tools in a Galaxy framework[33]. We chose Galaxy as our workflow system because the tools found in the ToolShed[34] have reached way beyond genome analysis for which Galaxy was originally developed. Just by focusing on the tool categories relevant to our study, one can cite proteomics, transcriptomics, metabolomics, flow cytometry analysis, and computational chemistry. Several communities are using Galaxy and many papers can be found online for omics (752 publications are found as of 16 February 2022) microbiome (380 publications), diseases like cancer (386 publications), and drug design and discovery (96 publications). At our request, a new Galaxy category named 'Synthetic Biology' was created, currently comprising 25 tools stored in the ToolShed.

The offering in Galaxy-SynBioCAD focuses on providing tools for pathway design and engineering. However, as Galaxy-SynBioCAD is a community effort, we anticipate our toolset will grow. Regarding

pathway design tools, many of the software products listed in the introduction could be considered to be added to the portal. In particular, strain design including knockout genes to maximize targeted product fluxes could easily be implemented via the flux balance analysis tools. Additionally, there are already Galaxy workflows to take up and analyze metabolomics flow cytometry data in the ToolShed[34], and these workflows could directly be incorporated into the portal to deal with data generated in the 'Test' step of the synthetic biology Design-Build-Test-Learn (DBTL) cycle. As mentioned in the introduction, several open-source software products deposited in GitHub[26–28,47] could address the 'Build' step and eventually provide drivers to automated constructions using different robotic workstations beyond those provided by Opentrons. Regarding the 'Learn' step in DBTL, the Opt-DoE tool could easily be adapted to propose new designs as it was done in Carbonell et al.[25]. Other approaches to be considered are methods that make use of active machine learning as in Borkowski et al.[48]. Although all design examples provided in the current paper are for engineering pathways in host organisms, because of the recent development of models (similar to genome scale models) for cell-free systems[49], one can also consider adapting the portal for design and engineering in cell-free.

All of the above-suggested additions could be implemented in our portal with relatively small efforts (cf. Tools design and integration process in 'Supplementary_Text' file). There are other applications that could be envisioned beyond pathway design and engineering. For instance, as shown in Delépine et al.[10] retrosynthesis software can easily be adapted to design biosensors. Such an adaptation has been proposed as a Galaxy-SynBioCAD workflow to predict and implement biosensors for the detection of various metabolites using the hydrogen peroxide compound as a metabolic hub, such that the metabolite signals are transduced to hydrogen peroxide before being sensed by the OxyR transcription factor[50]. Tools for genetic logic circuits engineering could also be considered. Also, as the cell resources consumption due to the level of expression of heterologous genes are not considered in the global score of a pathway, integrating in the overall design pipeline a tool enabling the prediction of the metabolic burden of genetic constructs based on enzyme expression levels would be of great interest and should be targeted for the future.

## Methods

### Retrosynthesis from target to chassis

Typically, the target compound, also named "source compound" is the compound of interest one wishes to produce, while the precursors are usually compounds that are natively present in a chassis strain. In the present implementation, the target can be any chemical that could be described by an InChI, and the chassis should be a metabolic model described in an SBML file. Starting from the source compound at the first iteration, reaction rules matching the chemical structure of the source are applied and newly predicted chemicals are generated. Reaction rules are generic descriptions of (bio)chemical reactions encoded into the community standard SMARTS[51]. The use of reaction rules allows estimating the outcomes of chemical transformations based on the generalization of reactions available in knowledge databases such as BRENDA[52], MetaCyC[52], Rhea[53], or MetaNetX[54]. The degree of generalization is controlled by describing the surrounding environment of the reaction center up to a given diameter as described in Duigou et al.[14]. To ensure the accuracy of the predicted transformations that will outcome from the reaction rules, the RetroRules dataset provided by the Galaxy RetroRules tool has been validated by (i) checking that rules allow to reproduce the template reactions, and (ii) checking that results obtained by decreasing diameters are supersets of results obtained with higher diameters. Only the reaction rules that successfully passed the 2 checks are retained. The validation of this dataset has a success rate of 99.3%.

For each reaction rule, a score is calculated based on the ability to retrieve enzyme sequences catalyzing substrate to product transformations, the method is detailed in Delépine et al.[10]. Newly produced chemicals are scanned and kept for the next iteration if they are not within the set of available precursors. In that way, a new iteration is started using the previously collected chemicals as the new source set. The iterative process stops when either no new chemicals are discovered or the predefined number of steps is reached. RetroPath2.0 carries out this task.

The retrosynthesis tools RetroPath2.0 and RP2Paths output a set of pathways composed by chemical transformations based on reaction rules. To obtain reactions, we have to re-build them from template reactions which have been used to generate the rules. In addition, within a pathway one single chemical transformation can reference multiple rules. Such pathways will be called master pathways. For each master pathway, the algorithm takes each transformation and creates one fork per reaction rule referenced. Then for each reaction rule, again the algorithm creates a new fork per template reaction used to build the current rule. The enumeration of all forks create a set of slightly different pathways (made of chemical reactions) for one master pathway. To perform the enumeration, datasets from RetroRules and MetaNetX are used. The reaction completion tool (rpCompletion) takes as input the CSV outputs of RP2paths and RetroPath2.0 and produces a collection of annotated SBML files. Those SBML files are "enriched" with additional information that are not stored as part of the normal SBML schema (cf. Pathway annotation section).

### Pathway annotation

Some results generated by the workflow produced in this study cannot be readily stored in the SBML files natively (informations about chemical reactions and species, thermodynamic and fluxes properties, as well as pathway information).

Using the Minimal Information Required In the Annotation of Models (MIRIAM) conventions, one can store within the SBML file information that instructs the user on the provenance of the reactions and chemical species within the model by cross-references to a wide range of databases. However, this needs a third-party database lookup to match the database ID with its structural information. In this work, intermediate products are generated ad hoc references and may not necessarily have database entries whereas some other information cannot be contained under the MIRIAM annotations as it is: SMILES, InChI, InChIKey for chemical species, reaction rule ID (RetroRules ID), associated template reaction ID, and rule score based on the expected enzyme availability.

As such, we elected to enrich the SBML format in such a way that our information can be stored directly within the SBML file without breaking any standard of the original file. Because SBML files are based on XML, new XML annotations are created outside the standard scope of an SBML file and thus are ignored by any standard SBML readers[55]. As a result, this enriched file format (denoted rpSBML) is fully compliant with SBML version 3 specifications.

Standard SBML extensions are also used in this project. The "groups" package is used to link the heterologous reactions and chemical species to identify them easily, as well as classifying the chemical species that are main actors in an heterologous pathway[55]. The SBML FBA package is used to define the FBA simulation conditions[56]. The tools also adhere to the MIRIAM annotation standard for the cross-references of chemical species to public databases[57].

### Flux balance analysis with fraction of reaction

We need metrics to rank heterologous pathways, this is why we developed an in-house Flux Balance Analysis (FBA) objective to simulate the flux of a target while considering the burden that the production of the target would cause on the cell. Under such simulation conditions, the analysis that returns a low flux may be caused by the starting native compound itself not having a high flux, or the cofactors required having a low flux, while the pathways with high flux would be caused by both the starting compound and the cofactors being in abundance. In either case, bottlenecks that limit the flux of the pathway may be identified and pathways that do not theoretically generate high yields can be filtered out. Furthermore, the production of heterologous molecules in an organism often causes a burden on the growth of the cell. To emulate such a condition, we use the method named 'fraction of reaction'. We first perform FBA (with COBRApy[58]) for the biomass reaction and record its flux. The upper and lower bounds of the biomass reaction are then set to the same amount, defined as a fraction of its previously recorded optimum. This ensures that any further FBA solution would have a fixed biomass production regardless of the conditions set for further analysis.

The tool optimizes the target molecule and records the flux directly to the SBML file and all changed bounds are reset to their original values before saving the file. It is important to note that orphan chemical species (those which are only consumed or produced by the metabolic pathway) are ignored. Such species are documented in the SBML file within the group named *rp_fba_ignored_species*.

### Thermodynamics

Thermodynamics is critical in synthetic pathway design by providing quantitative indicators to determine best metabolic pathways among a set of predicted ones. Thus, one can perform thermodynamic analysis to know whether a reaction direction of a pathway is feasible in physiological conditions.

In this work, we performed thermodynamic analysis for species, reactions and pathways. We use eQuilibrator[59] to compute the formation energy of chemical species and the Gibbs free energy for each chemical reaction and the heterologous pathway.

For each species involved in a heterologous pathway, the first challenge is to find the corresponding compound in the eQuilibrator database. To find the right compound, we try to exactly match species ID, InChIKey, InChI or SMILES and stop with the first hit. Then, if no compound is found, in the last resort, the first part of species InChIKey is looked for within the eQuilibrator cache and when the result (a list) is not empty, the first compound is taken. If species have no known structure neither in eQuilibrator database nor in any public one, the user has the possibility to specify substitution for identifier, InChI and InChIKey for these species. This substitution is documented in the SBML file with the group named *rp_thermo_substituted_species*. If a species has no known structure and is not substituted, then the reaction which involves this species will not have a thermodynamics value. Conversely thermodynamics can be computed by eQuilibrator for all reactions for which all species have been identified.

At the level of the pathway, we build a global pseudo-reaction linking chassis substrates to the target molecule and we compute thermodynamics with the eQuilibrator engine for the global pseudo-reaction.

Building the global pseudo-reaction requires finding the appropriate stoichiometric coefficients such that the intermediate compounds of the pathway cancel out. A linear optimization program (Eq. 1) can be set to find the stoichiometric coefficients. The program can be solved using SciPy[60] with a simplex algorithm.

$$
\begin{aligned}
\max \quad & c^T x \\
such\,that \quad & Ax = 0 \\
and \quad & 1 \leq x
\end{aligned}
\tag{1}
$$

where $c$ is the objective function, $A$ the stoichiometric matrix, and $x$ the unknown stoichiometric coefficient multipliers.

**Table 1 | Labware IDs used at Imperial College (London) and Micalis Institute (Paris) laboratories**

| Description | London | Paris | Used in |
|---|---|---|---|
| P20 single channel pipette | p20_single_gen2 | p20_single_gen2 | Steps 1, 3, and 4 |
| P300 multi channels pipette | p300_multi_gen2 | p300_multi_gen2 | Steps 2 and 4 |
| Opentrons 4-in-1 tubes rack for 1.5 ml eppendorf tubes | e14151500starlab_24_tuberack_1500ul | opentrons_24_tuberack_eppendorf_1.5ml_safelock_snapcap | Steps 1, 3, and 4 |
| Opentrons 10 μL tips rack | opentrons_96_tiprack_20ul | tipone_3dprinted_96_tiprack_20ul | Steps 1, 3, and 4 |
| Opentrons 300 μL tips rack | opentrons_96_tiprack_300ul | tipone_yellow_3dprinted_96_tiprack_300ul | Steps 2 and 4 |
| 96-well rigid PCR plate (clip reactions and transformation steps) | 4ti0960rig_96_wellplate_200ul | green_96_wellplate_200ul_pcr | Steps 1 and 4 |
| 96-well rigid PCR plate (purification and assembly steps) | 4ti0960rig_96_wellplate_200ul | black_96_wellplate_200ul_pcr | Steps 2 and 3 |
| Agar plate (transformation step) | nuncomnitraysingle_1_wellplate_35000ul corning_12_wellplate_6.9ml_flat | thermoomnitrayfor96spots_96_wellplate_50ul | Step 4 |
| Reservoir plate 21 mL 12 channels | 4ti0131_12_reservoir_21000ul | citadel_12_wellplate_22000ul | Step 2 |
| 96 deep well plate 2 mL wells | 4ti0136_96_wellplate_2200ul | transparent_96_wellplate_2ml_deep | Step 2 |

As an example, let's consider the following 3-reaction set:

$$Rxn_1 : MNXM188 + MNXM4 + MNXM6 + 3\,MNXM1$$
$$\rightarrow CMPD4 + CMPD3 + MNXM13 + MNXM15 + MNXM5$$

$$Rxn_2 : MNXM4 + 2\,CMPD3 \rightarrow 2\,MNXM1 + TARGET$$

$$Rxn_3 : MNXM4 + MNXM6 + 3\,CMPD4 \rightarrow MNXM13 + MNXM5$$

where *MNXM* are species IDs from MetaNetX, CMPD are intermediate species within the heterologous pathway and not present in the chassis organism, and TARGET is the product of interest. We note that $Rxn_2$ is the reaction to retain and *CMPD3* and *CMPD4* are species to remove as intermediate compounds. Thus, the parameters of linear solver are:

$$\begin{array}{ccc} R_1 & R_2 & R_3 \end{array}$$
$$c = \begin{pmatrix} 0 & 1 & 0 \end{pmatrix}$$

$$A = \begin{array}{c} \\ CMPD3 \\ CMPD4 \end{array} \begin{array}{ccc} R_1 & R_2 & R_3 \\ \begin{bmatrix} 1 & -2 & 0 \\ 1 & 0 & -3 \end{bmatrix} \end{array}$$

The solver outputs the following coefficients of reactions:

$$\begin{array}{ccc} R_1 & R_2 & R_3 \end{array}$$
$$x = \begin{pmatrix} 3 & 1.5 & 1 \end{pmatrix}$$

The global pseudo-reaction for the reaction sets becomes:

$$7.5\,MNXM1 + 3\,MNXM188 + 5.5\,MNXM4 + 4\,MNXM6$$
$$\rightarrow 4\,MNXM13 + 3\,MNXM15 + 4\,MNXM5 + TARGET$$

**Genetic design and engineering workflow execution**
From amongst the pathway predicted by the Pathway analysis workflow, the top-ranked one was selected with a score of 0.989. The search scope of the Selenzyme tool was restricted to the taxon ID of *Pantoea ananas*, i.e., 553 taxon ID. The combination of polycistronic constructs was built using 2 constitutive promoters (PJ23105 and PJ23116), 2 RBS linkers (A03 with a translation initiation rate of 46%, and A04 of 3%), 1

**Table 2 | DNA-Bot parameters that differ between Imperial College (London) and Micalis Institute (Paris) laboratories**

| Step | Parameter | London | Paris |
|---|---|---|---|
| Purification step | magdeck_id | magdeck | magnetic module gen2 |
| | magdeck_height | 20 | 10.8 |
| | settling_time | 2 | 6 |
| | drying_time | 5 | 15 |
| | elution_time | 2 | 5 |
| | wash_time | 0.5 | 0.5 |
| | bead_ratio | 1.8 | 1.8 |
| | incubation_time | 5 | 5 |
| Transformation step | incubation_temp | 4 | 8 |
| | incubation_time | 20 | 30 |

backbone (BASIC_SEVA_36_CmR-p15A.1) and enabling CDS permutation, resulting in a theoretical maximal number of constructs of 96 for 3 CDS. The labware IDs and parameters used with DNA-Bot parameters are listed in Tables 1 and 2. Additional changes in the purification step were needed because the 2 labs own different versions of the magnetic module (generation 1 vs generation 2). An updated version of the original DNA-Bot tool[47] was developed to be fully compatible with the Opentrons APIv2 and enabling a command-line interface, whilst retaining the option of using an enhanced GUI for direct user control.

**Lycopene production materials and methods**
Lycopene genes were synthesized by Twist Bioscience, flanked by the LMP prefix and the LMS suffix sequences[40], and cloned into pTwist high copy vector (AmpR, ColE1 replication origin) using Golden Gate. The resulting storage plasmids (pTwist_High_BASIC_CrtE, pTwist_High_BASIC_CrtB, pTwist_High_BASIC_CrtI) were confirmed by sequencing.

Storage plasmids for lycopene genes and assembly vector BASIC_SEVA_36_CmRp15A.1 were prepared using Monarch® Plasmid Miniprep Kit (Micalis) and E.Z.N.A.® Plasmid DNA Mini Kit (Imperial). The samples were diluted to 200 ng/μl ready to use in the clip reactions. Plasmids coding for the lycopene pathway variants were constructed using Biopart Assembly Standard Idempotent Cloning (BASIC) method. Five-part BASIC reactions were performed, replacing the dropout mScarlet cassette in the assembly vector

**Table 3 | Pathway and reaction features used by the XGBoost classifier**

|  | Item | Format | Comment |
|---|---|---|---|
| Pathway level | Chassis organisms | Integer | Taxonomy ID of the organism |
|  | Gibbs free energy | Float | Computed using the Thermodynamics calculations described in the Thermodynamics section |
|  | Fraction of reaction FBA | Float | Target flux computed by FBA (cf. Flux Balance Analysis with fraction of reaction section) |
| Reaction level | Reaction | 4096-bit vector | A reaction is represented by its Morgan fingerprint. Fingerprint(reaction) = Fingerprint(substrate) + Fingerprint(product). Morgan fingerprints are computed using the RDKit library[60]. |
|  | Enzyme availability score | Float | Enzyme availability score which provided a confidence level of finding an enzyme sequence catalyzing the reaction (cf. Delépine et al.[10] for details on score computation) |
|  | Gibbs free energy | Float | Computed using the Thermodynamics calculations mentioned above for the provided reaction only |

(BASIC_SEVA_36_CmRp15A.1) by a promoter and three genes with appropriate linkers. A collection of neutral and functional linkers (encoding RBS sequences) is available in a ready to use 96-well plate format (www.biolegio.com). For this work the standard BASIC linker set (Biolegio: BBP-18500) was used.

DNA-BOT was executed as described in detail in the Genetic design and engineering workflow execution section. A clip reaction master mix was prepared by combining 3 μL of 10X NEB T4 DNA ligase buffer, 1 μL NEB BsaI-HF v2 (NEB #R3733), 1 μL T4 DNA Ligase (NEB M0202 (Micalis), or Promega M1804 (Imperial)) per 20 μL required for each reaction. The Opentrons OT-2 pipetted the 20 μL master mix for each reaction, plus 1 μL of Biolegio BASIC linkers and 1 μL of each DNA parts, together with sufficient $H_2O$ to give a total volume of 30 μL. Clip reactions were incubated in a thermocycler (Applied Biosystems) for 30 cycles (37 °C for 5 min, 16 °C for 5 min), followed by a 5 min incubation at 60 °C at Micalis. Clip reactions were incubated in Opentrons Thermocycler Module for 20 cycles (37 °C for 2 min, 20 °C for 1 min), followed by a 10 min incubation at 60 °C at Imperial. For clip reaction purification, 54 μL of Mag-Bind® TotalPure NGS magnetic beads (OMEGA BIO-TEK (Micalis) or AMPure XP (Imperial)) were added; 150 μl 70% ethanol was used during wash steps; following re-suspension in $H_2O$, 40 μL of the eluent was transferred to a fresh well. Constructs were assembled in volumes of 15 μL using 1.5 μL of each purified clip reaction in a solution of 1X assembly buffer (CutSmart Buffer, NEB #B6004). Assembly reactions were incubated at 50 °C for 45 min in a thermocycler (Applied Biosystems (Micalis) or Opentrons Thermocycler Module (Imperial)). 20 μL of DH5-alpha Competent *E. coli* (NEB #C2987H, Micalis) or home-made DH5-alpha competent *E. coli* (Imperial) were distributed per well into 96-well plates; then, they were used for transformation reactions. In all, 5 μL of the assembly reactions were mixed with cells. Heat shock was conducted according to the manufacturer's instructions. SOC media (125 μL) was transferred to each assembly and the reaction incubated at 37 °C for 1 h with lids off. Transformation reactions were spotted on plates (Thermo Scientific™ OmniTray™ Single well) each containing 40 mL LB-agar supplemented with 17.5 μg/mL chloramphenicol. The spotting protocol was run twice in order to spot 2 times 5 μL for each transformation reaction. The spotting step at Imperial was repeated using 40 μL of transformation reaction on a 12-well plate (Costar® 12-well 3737), each well containing 10 mL of LB_agar supplemented with 17.5 μg/mL chloramphenicol. 100 μL of each transformation reaction was plated manually on LB-agar plates containing 17.5 μg/mL chloramphenicol as well.

We selected 6 colonies at Micalis, and 8 colonies at Imperial, with different levels of red color (visual inspection) and sequenced them. This was followed by lycopene extraction from the 4 and 8 colonies, respectively, that sequenced correctly. To quantify lycopene production, 2 mL of overnight cultures grown in LB (Cm 17.5 μg/mL) were pelleted at 5000×*g* (10 min), washed by re-suspension in 1 mL water, re-pelleted at 5000×*g* (10 min), and the pellet re-suspended for extraction in 1 mL acetone. The cells in acetone were incubated at 55 °C for 20 min with continuous shaking (1300 rpm, Eppendorf Thermomixer comfort), centrifuged at 19000 x g (10 min), and the supernatant

transferred to a fresh tube. Lycopene absorbance of the supernatant was measured at 474 nm using a quartz cuvette (Hellma 104.002B-QS) in a spectrophotometer (UVisco V-1100D (Micalis) or NanoDrop™ One UV-Vis (Imperial)), and the pellet was dried at 50 °C for 48 h to determine the gDCW. Absorbance ($OD_{474}$) was converted to molar concentration value by dividing by 150479, the molar extinction coefficient (ε) of lycopene[61]. The yield per gram dry cell weight (mg/gDCW) was calculated by dividing the absolute yield (mg) by the weight of the dried cell pellet.

**Machine learning global scoring**

The purpose of the machine learning model is to predict if a given predicted pathway is a valid one or not. To that end, we developed a classifier based on the XGBoost library[62]. The classifier was trained on 7919 Galaxy-SynBioCAD generated pathways (comprising 43392 reactions) used during the expert validation trial where 754 pathways were labeled positive. The training set can be found in the Supplementary file 'Dataset 3'. The input features used by the classifier are given in the Table 3, these were computed for each training pathway and each reaction within the pathways. XGBoost learn function was parameterized with a Maximum depth of a tree of 1000 and a step size shrinkage used in update to prevent overfitting of 0.3 (default value). The classifier accuracies were recorded during 4-fold cross validation.

**Reporting summary**

Further information on research design is available in the Nature Research Reporting Summary linked to this article.

## Data availability

Source data of figures are provided in the Source Data file. The eQuilibrator database is available online at Zenodo (https://doi.org/10.5281/zenodo.4128543) and it can be queried using the equilibrator-api python library which is available online at github (https://gitlab.com/equilibrator/equilibrator-api). The LASER database is available online at bitbucket (https://bitbucket.org/jdwinkler/laser_release/src/master/). All other relevant data are included in the paper and in the Supplementary Text and Dataset files. Source data are provided with this paper.

## Code availability

All codes are accessible online following links provided in Table S1 found the Supplementary Text file.

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

## Acknowledgements

M.d.L. and J.L.F. acknowledge funding provided by the infrastructure IBISBA1.0 (Horizon 2020 under grant agreement No 730976). J.H., T.D., K.B.K., and J.L.F. funding provided by the project ALADIN (ANR EQUIPEX+). T.D., J.L.F., and G.S.B. funding provided by BioRoBoost (Horizon 2020 under Grant agreement No 820699) and PC, NS and JLF funding from the Biotechnology and Biological Sciences Research Council (BBSRC) and the Engineering and Physical Sciences Research Council (EPSRC) under grant 'Centre for synthetic biology of fine and specialty chemicals (SYNBIO-CHEM)' (BB/M017702/1). N.S. acknowledges further funding from the BBSRC under grant 'GeneORator: a novel and high-throughput method for the synthetic biology-based improvement of any enzyme' (BB/S004955/1) and from the University of Liverpool. PC also acknowledges support from the Universitat Politècnica de València Talento Programme. VZ was supported for this work by The Edinburgh Genome Foundry funded by the BBSRC (BB/M025659/1, BB/M025640/1, and BB/M00029X/1 to Susan Rosser) and the BBSRC/MRC/EPSRC funded UK Centre for Mammalian Synthetic Biology (BB/M0101804/1 to Susan Rosser) as part of the RCUK's Synthetic Biology for Growth programme. M.K. acknowledges funding support from INRAe's MICA department, U. Paris-Saclay, Ile-de-France (IdF) region's DIM-RFSI, and ANR DREAMY (ANR-21-CE48-003). G.S.B. and G.B. acknowledge the support of the EPSRC (EP/R034915/1). We thank Thomas Nowak (LISN, Université Paris-Saclay) for the use of the Opentrons OT-2 robot (funded by project ABIDE, CNRS), Bikash Ranjan Samal and Bibal Shahin (Micalis Institute, U. Paris Saclay) for their preliminary work of FBA and machine learning for pathway scoring. We also thank the validation trial expert panel: PhD students and master students in System and Synthetic Biology at the University of Paris Saclay (Angelo Cardoso-Batista, Léon Faure, Mostafa Mahdy, Paul Soudier and Myint Toe), the Imperial College 2020 iGEM team, the University of Manchester SYNBIOCHEM centre (in particular Rosalind Le Feuvre who dispatched the batches), Muriel Gondry (I2BC, U. Paris Saclay), Ioana Popescu (Genoscope, U. Paris Saclay), Amit Pathania and Forum Shah (Micalis Institute, U. Paris Saclay), Jacques Haiech (U. Strasbourg), Fayza Daboussi, Stéphanie Heux, Thomas Lautier, Pierre Millard, and Gilles Truan (TBI, INSA-Toulouse), Baudoin Délepine, Remi Peyraud, and Jonathan Verbeke (iMEAN Biotech), and Joan Albiol (Universitat Autònoma, Barcelona).

## Author contributions

J.H., T.D., and J.L.F. designed the study and wrote the main text of the paper. J.H. and T.D. supervised code development and packaging for in-house tools. K.B.K. packaged tools and published them on Galaxy ToolShed. J.H. performed the deployment of all tools available in the Galaxy-SynBioCAD web portal as well as the Galaxy platform (web, db, tools, networking). M.d.L. encoded preliminary versions of Galaxy-SynBioCAD tools, and collected data for the literature benchmarking. OT developed the rpReport tool. N.S., P.C., T.G., and Z.A. integrated several tools in the portal with the help of M.d.L. and wrote the corresponding description. G.B., G.S.B., M.K., M.S.A., T.D., and Y.E.M. worked on the updated version of DNA-Bot. G.B., M.K., M.S.A., and Y.E.M. worked on the Experimental Benchmarking, and M.K. and M.S.A. wrote the relevant section. All authors read and approved the manuscript.

## Competing interests

The authors declare no competing interests.
