## [Peer Review File · Nature Communications]

The Automated Galaxy-SynBioCAD Pipeline for Synthetic Biology Design and EngineeringREVIEWER COMMENTS

Reviewer #1 (Remarks to the Author):

This paper introduces a Galaxy-SynBioCAD portal as toolshed for synthetic biology. The tools and workflows currently shared on the portal enables one to build libraries of strains producing desired chemical targets covering an end-to-end metabolic pathway design and engineering process from the selection of strains and targets, the design of DNA parts to be assembled, to the generation of scripts driving liquid handlers for plasmid assembly and strain transformations. Standard formats like SBML and SBOL

are used throughout to enforce the compatibility of the tools.

Authors illustrated the link between pathway design and engineering with the building of a library of E. coli lycopene-producing strains. They found an 83% success rate in retrieving the validated pathways among the top 10 pathways generated by the workflows. Following concerns should be addressed:

1. The Galaxy-SynBioCAD portal seems only suitable for E. coli. If that is the case, please state it. Please discuss its possibility to extend to other non-model organisms;
2. Lycopene synthesis is encoded by only five genes. If more than five genes pathways are involved, would the Galaxy-SynBioCAD portal require modification?
3. Page 10, next to the bottom paragraph "Using the literature and expert validated pathways, we developed a machine learning model whose purpose was to evaluate if any given pathway is a valid one or not. To that end, we used a classifier (cf. Machine Learning Global Scoring in Methods), which returned scores for the queried pathways. The score distribution for the dataset of 7919 pathways is given in Figure 2.G (raw data are in Supplementary file 'ML_Scores'). The classifier exhibited excellent performances with an average cross-validation accuracy of 0.91.". Authors should state if the 7919 pathways are all encoded in E. coli? or any other organisms?
4. Could authors demonstrate more application of the Galaxy-SynBioCAD portal for other products? not just one?

Reviewer #2 (Remarks to the Author):

This paper presents a new portal and proposed a means in interfacing individual software already developed for synthetic biology to achieve more complete workflows related to the design and engineering of engineered microbes. Interoperability among the software developed for synthetic biology has been a challenge. The work presented was inspired by scientific workflows, particularly the Galaxy system which now has over 8000 tools that are made available via public ToolShed. The authors proposed taking similar approach as the Galaxy system to create a portal comprising tools more directed to Synthetic Biology, Metabolic Engineering and industrial biotechnology. Hence the authors named it as Galaxy-SynBioCAD.

In the paper, the authors demonstrated the utility of the Galaxy-SynBioCAD from design to engineering workflow by building a library of lycopene producing E. coli. At the same time, the authors demonstrated that the workflow can be implemented at different sites.

I find the most important aspect of the work is the demonstration of interoperability among the software tools since the individual software has already been published earlier and using this new approach to demonstrate the design and engineering of the microbes. Overall, the paper is well written. I find the paper interesting and presents useful advancement to the field, by offering a means to enable different software to be used in a workflow manner through the use of standard formats such as SBML and SBOL and also providing a good demonstration of the utility of the created

workflows with experiments. However, I have a few questions as follows:

1. For the selection of the software to be included in the integrated, the authors mentioned that the software should be using standard input/output. What would the standard input/output be?
2. "Of the 88 constructs, this time we obtained transformants for 51 (41 red + 10 white) and 63 (49 red + 14 white) constructs at Paris and Imperial, respectively, including 36 (33 red + 3 white) constructs in common". Looks like quite a number of constructs failed. The reasoning of the failed constructs is attributed to potential burden caused by the constructs designs. However, my understanding is that the Pathway Analysis workflow considers "the burden that the production of the target would cause on the cell". Was it inadequate?
3. Among the constructs that form colonies, how did the labs go about choosing the 4 constructs (Paris) and the 8 constructs (London) to characterize the production of lycopene? And did the performance of these characterized constructs in terms of lycopene production agree with what was predicted by the Pathway Analysis workflow?
4. How was the production of lycopene compared to what was published in the literature?
5. In the section "Expert validation trial benchmarking", can the authors clarify how the success rate 83% was computed?
6. Was the machine learning classifier meant as a way to validate/benchmark the pathways generated by Galaxy-SynBioCAD workflows or was it meant to be used to score and rank top valid pathways generated by Galaxy-SynBioCAD? Would the top 10/50 pathways scored by Galaxy-SynBioCAD without the classifier consist of literature/expert validated pathway?
7. Page 10, "we used a classifier (cf. Machine Learning Global Scoring in Methods), which returned scores for the queried pathways." It wasn't clear what the score is. Was this score referring to the matching score described in the supplementary "Literature Pathways matching algorithm"? Or is the classifier generating a different score? And is the matching score an input feature to the classifier?
8. Under "Machine Learning Global Scoring", I can't seem to find supplementary file 'ML_Scores' mentioned in the following sentence: "The training set can be found in the Supplementary file 'ML_Sores'."
9. In Discussion, can the authors comment on the current limitations of Galaxy-SynBioCAD and discuss key learning points about the integration of the new software into Galaxy-SynBioCAD? I believe it will be useful for the community who would like to contribute to the effort.
10. As the tools selected are already published, what are the challenges faced in interfacing the different software? Was the current SBOL and SBML standard adequate to support the interfacing?
11. In Discussion, can the authors provide suggestions on the key considerations for developers who are interested to contribute their software to the Toolshed?

We thank the reviewers for their constructive comments we believe improve our manuscript. Below we address point-by-point the questions that were raised (under ANSWER) and also list the modifications we made to the manuscript (under ACTION)

Reviewer #1 (Remarks to the Author):

This paper introduces a Galaxy-SynBioCAD portal as toolshed for synthetic biology. The tools and workflows currently shared on the portal enables one to build libraries of strains producing desired chemical targets covering an end-to-end metabolic pathway design and engineering process from the selection of strains and targets, the design of DNA parts to be assembled, to the generation of scripts driving liquid handlers for plasmid assembly and strain transformations. Standard formats like SBML and SBOL are used throughout to enforce the compatibility of the tools. Authors illustrated the link between pathway design and engineering with the building of a library of *E. coli* lycopene-producing strains. They found an 83% success rate in retrieving the validated pathways among the top 10 pathways generated by the workflows. Following concerns should be addressed:

1. The Galaxy-SynBioCAD portal seems only suitable for *E. coli*. If that is the case, please state it. Please discuss its possibility to extend to other non-model organisms

ANSWER

Although we demonstrate the use of the Galaxy-SynBioCAD portal for metabolic engineering in *E. coli*, there are 47 organisms on Galaxy-SynBioCAD directly available through a list in the RetroSynthesis workflow (*cf.* Figure S4 below). In addition, a user can import any model of his/her choice in SBML format. For instance, models imported by users can come from the BiGG models database (<http://bigg.ucsd.edu/>).

ACTION

We added the following sentences in the main manuscript (Methods > RetroSynthesis from target to chassis, added text is in **bold**):

Typically, the target compound, also named “source compound”, is the compound of interest that one wishes to produce, while the precursors are usually compounds that are natively present in a chassis strain. **In the present implementation, the target can be any chemical that could be described by an InChI, and the chassis should be a metabolic model described in an SBML file. [...]**

We also added Figure S4 in the Supplementary Text (section Retrosynthesis workflow):

Lastly, rpCompletion takes those individual metabolic pathways to filter them (duplicated pathways are removed), then splits them into sub-pathways by adding the appropriate cofactors, and finally converted them to SBML files. Additional details are provided in the Methods section (*cf.* Pathway completion combinatorics). **The Figure S4 is a screenshot of the Retrosynthesis workflow as it appears in Galaxy.**

Workflow: RetroSynthesis ✓ Run Workflow

History Options

Send results to a new history
 No

1: Target to produce

Target to produce

2: Pick SBML Model (Galaxy Version 0.0.1)

Strain

- ✓ Acinetobacter baumannii AYE (iCN718)
- Bacillus subtilis subsp. subtilis str. 168 (iYO844)
- Chlamydomonas reinhardtii (iRC1080)
- Clostridioides difficile 630 (iCN900)
- Clostridium ljungdahlii DSM 13528 (iHN637)
- Cricetulus griseus (iCHOv1_DG44)
- Cricetulus griseus (iCHOv1)
- Escherichia coli str. K-12 substr. MG1655 (iAF1260b)
- Escherichia coli str. K-12 substr. MG1655 (iAF1260)
- Escherichia coli str. K-12 substr. MG1655 (iML1515)
- Escherichia coli str. K-12 substr. MG1655 (iJO1366)
- Escherichia coli str. K-12 substr. MG1655 (iJR904)
- Escherichia coli str. K-12 substr. MG1655 (e_coli_core)
- Geobacter metallireducens GS-15 (iAF987)
- Helicobacter pylori 26695 (iT341)
- Homo sapiens (iAT_PLT_636)
- Homo sapiens (Recon3D)
- Homo sapiens (iAB_RBC_283)
- Homo sapiens (RECON1)
- Klebsiella pneumoniae subsp. pneumoniae MGH 78578 (iYL1228)
- Lactococcus lactis subsp. cremoris MG1363 (iNF517)
- Methanosarcina barkeri str. Fusaro (iAF692)
- Mus musculus (iMM1415)
- Mycobacterium tuberculosis H37Rv (iNJ661)

Figure S4. Screen caption of the Retrosynthesis workflow.

The workflow configuration panel highlights the selection of the chassis model from a predefined list. Users can also import models in SBML file format.

2. Lycopene synthesis is encoded by only five genes. If more than five genes pathways are involved, would the Galaxy-SynBioCAD portal require modification?

ANSWER

The Galaxy-SynBioCAD portal represents the “one-stop shop”, i.e., a “unified way” to access collections of tools geared toward Synthetic Biology and Engineering applications. The portal can be applied to pathways longer than five genes without any modifications.

Lycopene predicted pathways shown in the manuscript involved 3 genes that encode the 3 metabolic steps required for the production of lycopene by *E. coli*. To predict longer pathways, the parameters used at execution time should be specified for a few tools, namely for RetroPath2.0 (“Maximal Pathway length” parameter) and BasicDesign (“Maximum Gene per Construct” parameter). However, no modification of the tools (i.e. the source code) is required.

ACTION

At the time of answering this question, we found out that the “Maximum Gene per Construct” parameter in the BasicDesign tool could not be changed by users (only the default value of 3 was possible). We therefore made the appropriate changes so that the parameter can now be set either

through the Galaxy interface or using the python Command Line Interface. The BasicDesign code has been updated on GitHub (<https://github.com/brsynth/rpbasicdesign/>), corresponding package and the Galaxy node has been updated on Anaconda Cloud (<https://anaconda.org/conda-forge/rpbasicdesign>), the Galaxy ToolShed (<https://toolshed.g2.bx.psu.edu/view/tduigou/rpbasicdesign/9ba4dab7f0ba>) and Galaxy-SynBioCAD.

3. Page 10, next to the bottom paragraph "Using the literature and expert validated pathways, we developed a machine learning model whose purpose was to evaluate if any given pathway is a valid one or not. To that end, we used a classifier (cf. Machine Learning Global Scoring in Methods), which returned scores for the queried pathways. The score distribution for the dataset of 7919 pathways is given in Figure 2.G (raw data are in Supplementary file 'ML_Scores'). The classifier exhibited excellent performances with an average cross-validation accuracy of 0.91.". Authors should state if the 7919 pathways are all encoded in *E. coli*? or any other organisms?

ANSWER

The 7919 pathways used by the classifier are all linked to 4 chassis organisms, namely *E. coli*, *S. cerevisiae*, *P. putida* and *B. subtilis*. This comprise pathways extracted from literature coming from 3 organisms (*E. coli*, *S. cerevisiae*, and *P. putida* as shown in Fig. 3) to which additional pathways from the Laser database were added (including pathways from *B. subtilis*).

ACTION

We added the above information in the manuscript (*Results > Expert validation trial benchmarking*, added text is in **bold**):

The set included the 5874 pathways generated for our literature benchmarking along with 2045 additional pathways corresponding to 20 additional (target, chassis) pairs taken from the Laser database, **which includes some pathways from *B. subtilis*.**

4. Could authors demonstrate more application of the Galaxy-SynBioCAD portal for other products? not just one?

ANSWER

We showed an end-to-end use case with the automated implementation of lycopene *E. coli* producing strains, involving the sequential execution of 3 workflows, namely Retrosynthesis, Pathway Analysis, and Genetic Design (BASIC).

Furthermore, we showed for 60 additional compounds how predicted pathways are correctly ranked compared to the pathways implemented in 3 organisms (namely *E. coli*, *S. cerevisiae*, *P. putida*) extracted from experimentally validated engineering constructs. The list of target compounds and the ranking of the predictions compared to the literature extracted pathways are summarized in Figure 3.

Overall, our workflows were applied to 60 targets and not just one. To go beyond metabolic engineering showcases, we introduce an additional application of the Galaxy-SynBioCAD in the context of biosensing design (see the manuscript change below).

ACTION

To supplement the metabolic engineering showcases, we mention in the Discussion section an additional application of the Galaxy-SynBioCAD in the context of biosensing design (addition in **bold**) :

All of the above-suggested additions could be implemented in our portal with relatively small efforts. There are other applications that could be envisioned beyond pathway design and engineering. For instance, as shown in Delepine *et al.*¹⁰ retrosynthesis software can easily be adapted to design

biosensors. Such an adaptation has been proposed as a Galaxy-SynBioCAD workflow to predict and implement biosensors for the detection of various metabolites using the hydrogen peroxide compound as a metabolic hub⁴⁸. Tools for genetic logic circuits engineering could also be considered.

Reviewer #2 (Remarks to the Author):

This paper presents a new portal and proposed a means in interfacing individual software already developed for synthetic biology to achieve more complete workflows related to the design and engineering of engineered microbes. Interoperability among the software developed for synthetic biology has been a challenge. The work presented was inspired by scientific workflows, particularly the Galaxy system which now has over 8000 tools that are made available via public ToolShed. The authors proposed taking similar approach as the Galaxy system to create a portal comprising tools more directed to Synthetic Biology, Metabolic Engineering and industrial biotechnology. Hence the authors named it as Galaxy-SynBioCAD.

In the paper, the authors demonstrated the utility of the Galaxy-SynBioCAD from design to engineering workflow by building a library of lycopene producing *E. coli*. At the same time, the authors demonstrated that the workflow can be implemented at different sites.

I find the most important aspect of the work is the demonstration of interoperability among the software tools since the individual software has already been published earlier and using this new approach to demonstrate the design and engineering of the microbes. Overall, the paper is well written. I find the paper interesting and presents useful advancement to the field, by offering a means to enable different software to be used in a workflow manner through the use of standard formats such as SBML and SBOL and also providing a good demonstration of the utility of the created workflows with experiments. However, I have a few questions as follows:

1. For the selection of the software to be included in the integrated, the authors mentioned that the software should be using standard input/output. What would the standard input/output be?

ANSWER

There are several standard file formats, some are very generic (CVS, TSV, JSON...) while others are more specific to a scientific field (e.g. SBOL, SBML). Within a workflow, each tool is connected to one or several other tools and then has to have a common file format to exchange data, i.e. each output file of a tool has to fit to the input file format to downstream tools in the workflow.

Currently, Galaxy-SynBioCAD hosts 21 tools and among these 19 are integrated within the 4 workflows presented in this article (RetroSynthesis, Pathway Analysis, Genetic Design (2 workflows)). All these tools now make use of standard I/O file formats. From amongst them, the SBML and SBOL formats are used whenever possible. All tools in the Pathway Analysis workflow (rpFBA, rpThermo, rpScore) as well as Selenzyme and rpBASiCDesign use SBML format because they deal with objects related to metabolic pathways (compounds, reactions, pathway, compartment...). The tools that are using SBOL format are: PartsGenie, OptDoE, DNAWeaver, LCRGenie.

Each time a new tool is integrated into the ecosystem, the best option is to fit its I/O format with upstream and downstream tools, according to where it would be plugged-in within a workflow. In some scenarios, it is easier using some additional I/O format for tools already present in Galaxy-SynBioCAD. Finally, if nothing above is feasible, it always remains the solution to use an already existing converter or develop a new one.

ACTION

We added the following text in the main text (introduction of the Results section, added text is in **bold**):

Within a workflow, each tool connected to one or more tools must share common file format for data exchange, i.e. each output file of a tool has to be compatible with the input file format of downstream tools in the workflow. The file format relies on the nature of the data (e.g. metabolic model, metabolic pathway, construct design) and the implementation choice made for each tool. Among the standard formats used, some are rather generic (CSV, TSV, JSON) while others are more specific to a scientific field (e.g. SBOL, SBML).

2. “Of the 88 constructs, this time we obtained transformants for 51 (41 red + 10 white) and 63 (49 red + 14 white) constructs at Paris and Imperial, respectively, including 36 (33 red + 3 white) constructs in common”. Looks like quite a number of constructs failed. The reasoning of the failed constructs is attributed to potential burden caused by the constructs designs. However, my understanding is that the Pathway Analysis workflow considers “the burden that the production of the target would cause on the cell”. Was it inadequate?

ANSWER

The Pathway Analysis workflow estimates the theoretical efficiency of pathways, notably based on the topology of the metabolic networks of both the chassis and the predicted pathways, the target production flux, the organism’s growth rate, and the thermodynamics of the reactions based on reaction rule scores which reflect the enzyme availability.

These criteria and a few others (like the pathway length and fingerprints of reactions) are combined into a final score we named “global score”. However, the cell resources consumption due to the level of expression of heterologous genes is not considered.

For the lycopene producing pathways, we predict 9 pathways using the Retrosynthesis workflow. We ranked those pathways using the Pathway Analysis workflow, and we selected the best one (having a global score of 0.98947) to engineer 88 constructs in *E coli* strains. Hence, the 88 constructs experimentally implemented shared the same global score.

Integrating in the overall design pipeline a tool enabling the prediction of the metabolic burden of genetic constructs based on enzyme expression levels would be of great interest and should be targeted for the future. However, as far as we know, such a publicly available tool has not yet been released.

ACTION

‘Pathway’ and ‘Construct’ are two concepts used within the Galaxy-SynBioCAD tools. The Pathway Analysis workflow ranks ‘Pathways’, not ‘Constructs.’ To ensure the distinction between the two, we added the following statement in the manuscript (section *Results > Pathway design and engineering tools and workflows*, addition in **bold**):

In the reminder of the paper, a metabolic pathway is a succession of chemical reactions transforming reactants into products, while a construct is the assembly of genetic parts that encodes a metabolic pathway. Several constructs with different enzyme and regulatory sequences can encode for the same pathway. The *Retrosynthesis* and *Pathways analysis* workflow generate annotated SBML files **describing pathways** (cf. **Pathway annotation** in **Methods**), which are taken as input to the *Genetic design and engineering* workflows to produce **constructs**, i.e. plasmids encoded in SBOL format along with assembly plans (in CSV files) and liquid handler instructions (Python scripts directly executable by Opentrons).

The following sentence was also added at the end of *Discussion* section:

Also, as the cell resources consumption due to the level of expression of heterologous genes are not considered in the global score of a pathway, integrating in the overall design pipeline a tool enabling the prediction of the metabolic burden of genetic constructs based on enzyme expression levels would be of great interest and should be targeted for the future.

3. Among the constructs that form colonies, how did the labs go about choosing the 4 constructs (Paris) and the 8 constructs (London) to characterize the production of lycopene? And did the performance of these characterized constructs in terms of lycopene production agree with what was predicted by the Pathway Analysis workflow?

ANSWER

The constructs were chosen based on a visual inspection of the colony color. We selected 6 colonies in Paris, and 8 colonies in London, with different levels of red color and sequenced them. This was followed by lycopene extraction from the 4 and 8 colonies, respectively, that sequenced correctly. The Pathway Analysis workflow ranks the enzyme combinations without taking into account the expression levels of the different enzymes (cf. answer question 2). Since we used the same genes for all our constructs (with varying levels of enzyme expression), they all had the same global score (0.98947) as calculated by the Pathway Analysis workflow.

ACTION

We have added the following text within *Methods > Lycopene production materials and methods* section (modifications are in bold):

We selected 6 colonies at Micalis, and 8 colonies at Imperial, with different levels of red color (visual inspection) and sequenced them. This was followed by lycopene extraction from the 4 and 8 colonies, respectively, that sequenced correctly. To quantify lycopene production, 2 mL of overnight cultures grown in LB...

4. How was the production of lycopene compared to what was published in the literature?

ANSWER

Several groups have expressed lycopene in different *E. coli* strains. For example, Kim *et al.*, 2001 (Keasling lab) overexpressed the upstream *dxs* enzyme to obtain a yield of 5.69 mg/gDCW lycopene from *E. coli* DH5-alpha cells grown in 2xYT medium (<http://www.ncbi.nlm.nih.gov/pubmed/11180061>). More recently, Sun *et al.*, 2014 obtained 6.52 mg/gDCW lycopene from a different *E. coli* strain (ATCC 8739) grown in LB medium, after several upstream genetic optimizations ($\Delta crtXY$, *idi* & *dxs* overexpression) (<https://pubmed.ncbi.nlm.nih.gov/24806808/>). In our work, we produce 4.4 mg/gDCW lycopene from DH5-alpha cells grown in LB medium, using only combinatorial optimization of *crtEBI* expression levels without any upstream genetic optimization. This is our highest yield obtained when all of the three genes are expressed from the weaker RBS A04 (Fig 4D).

ACTION

We have added the following text within *Results > Genetic design & engineering workflow benchmarking for lycopene production* section (modifications in bold):

The highest lycopene production was obtained for construct G6 (4.389 mg/gDCW), a yield comparable to those from *E. coli* in similar conditions (5.69 mg/gDCW from *E. coli* DH5-alpha in 2xYT⁴² and 6.52 mg/gDCW from *E. coli* ATCC 8739 in LB⁴³).

5. In the section “Expert validation trial benchmarking”, can the authors clarify how the success rate 83% was computed?

ANSWER

We selected 60 target molecules from literature that were engineered in different chassis organisms (*E. coli*, *S. cerevisiae* and *P. putida*). The *RetroSynthesis workflow* was then run for the same target/organism and the 113 literature or expert flagged pathways were identified using the matching algorithm described in the ‘Supplementary_Text’ file. It turned out that in 94/113 (83%) cases, the literature or expert selected pathways had a score in the top 10 scores of all the pathways generated for the same target and chassis. Results are presented in Fig. 3 and in the Supplementary Dataset 3 (tab Lit_Pathway_Rank_ML).

ACTIONS

We modified the text in bold below (in section *Results > Expert validation trial benchmarking*):

The machine learning scoring process was used to rank the top 50 Galaxy-SynBioCAD pathways generated for 60 target molecules taken from our literature pathway training set. **More precisely, a global score was calculated for all SynBioCAD generated pathways using machine learning. Concomitantly, literature or expert selected pathways were identified using the matching algorithm described in the ‘Supplementary_Text’ file. Results are shown in Figure 3 where each row is a ranked list of collections of SynBioCAD generated pathways for a given target molecule in a given chassis. Literature or expert selected pathways are flagged with a black square and pathways are ranked according to the global score aforementioned. Overall, we find that in 83% cases, the literature or expert selected pathways have a score in the top 10 scores of all the pathways generated for the same target and chassis. The number rises to 94% in retrieving literature or expert validated pathways among the top 50 machine learning scored pathways.**

6. Was the machine learning classifier meant as a way to validate/benchmark the pathways generated by Galaxy-SynBioCAD workflows or was it meant to be used to score and rank top valid pathways generated by Galaxy-SynBioCAD? Would the top 10/50 pathways scored by Galaxy-SynBioCAD without the classifier consist of literature/expert validated pathway?

ANSWER

The machine learning classifier was meant to be used to score and rank pathways generated by Galaxy-SynBioCAD. To answer the second part of the question (“*Would the top 10/50 pathways scored by Galaxy-SynBioCAD without the classifier consist of literature/expert validated pathway?*”), we computed a score without making use of ML. Briefly, we compiled for each generated pathway the inverse of the pathway length (favoring shorter pathway), the opposite normalized pathway free energy (favoring high negative values), the normalized averaged enzyme availability score and the normalized FBA calculated product flux value. All these values were calculated by the *Pathway analysis workflow*. A direct score calculation, which does not make use of ML, can simply be obtained summing the above parameters. Results are presented in the Supplementary Dataset 3 (tab Lit_Pathway_Rank_noML) where 58% of the identified literature/expert pathways are found within the top 10 scored pathways generated by Galaxy-SynBioCAD, this number is lower than the one obtained when using ML (83%).

ACTION

We added the text below at the end of section *Results > Expert validation trial benchmarking* (modifications in **bold**):

To assess the advantage of making use of a machine learning scoring schema, we computed a direct score based on (i) the inverse of the pathway length (favoring shorter pathway), (ii) the opposite

normalized pathway free energy (favoring high negative values), (iii) the normalized averaged enzyme availability score and (iv) the normalized FBA calculated product flux value. All these values were calculated using the *Pathway analysis workflow* (additional information are found in Table 3 in Methods). A direct score calculation, which does not make use of machine learning, can simply be obtained summing the above four parameters. Results are presented in the Supplementary file ‘Dataset 3’ (tab Lit_Pathway_Rank_noML) where 58% of the identified literature/expert pathways are found within the top 10 scored pathways generated by Galaxy-SynBioCAD, this number is lower than the one obtained when using machine learning (83%).

7. Page 10, “we used a classifier (cf. Machine Learning Global Scoring in Methods), which returned scores for the queried pathways.” It wasn’t clear what the score is. Was this score referring to the matching score described in the supplementary “Literature Pathways matching algorithm”? Or is the classifier generating a different score? And is the matching score an input feature to the classifier?

ANSWER

The ML score is different from the matching score. The matching score captures the similarity between a literature pathway and a pathway built by Galaxy-SynBioCAD (cf. Fig. 2D). The matching score is calculated by the algorithm mentioned by the Reviewer and described in the Supplementary text. The machine learning score is returned by the classifier (XGBoost). The matching score is used to set the positive and negative labels of the training set of the classifier. As shown in Fig 2.D, all pathways with a matching score > 0.5 are labeled positive.

ACTION

We modified the caption of Figure 2 as follows:

(D) The Galaxy-SynBioCAD generated pathways are compared with the literature pathways using a matching algorithm (cf. Supplementary_Text file). The plot shows for each literature pathway the best matching pathways among all Galaxy-SynBioCAD generated pathways. Pathways having a matching score above 0.5 are identical (similarity of 1) to literature pathways as far as main substrate and products are concerned. The raw data can be found in Supplementary file ‘Dataset 2’ (tab literature_matching_score)

(G) The set of labeled pathways is used to train a classifier printing out a machine learning score to assess if a given pathway is valid or not (cf. Machine Learning Global Scoring in Methods). The figure plots results obtained for all pathways generated by Galaxy-SynBioCAD. The raw data, including the training set, can be found in the Supplementary file ‘Dataset 3’. Using a machine learning global score threshold of 0.5, the accuracy retrieving literature of expert labeled pathways is 0.91 with a false positive rate of 0.10 in 4-fold cross validation (cf. Supplementary file ‘Dataset 3’, tab Pathway_PredictedScore).

8. Under “Machine Learning Global Scoring”, I can’t seem to find supplementary file ‘ML_Scores’ mentioned in the following sentence: “The training set can be found in the Supplementary file ‘ML_Scores’.”

ANSWER

We checked the file uploaded in the *Nature Communication* submission systems. There are 3 supplementary files but these have been renamed automatically by the submission system - the ML_Scores file was renamed Supplementary Dataset 3.

ACTION

In the revised version in the main text and supplementary text we renamed all supplementary files in the order they appear: Supplementary Dataset 1 (old name Lycopene_Benchmark) , Supplementary Dataset 2 (old name Literature_Pathways) and Supplementary Dataset 3 (old name ML_Scores).

9. In Discussion, can the authors comment on the current limitations of Galaxy-SynBioCAD and discuss key learning points about the integration of the new software into Galaxy-SynBioCAD? I believe it will be useful for the community who would like to contribute to the effort.

ANSWER

Since questions 9 and 11 are related, the following is a joint answer spanning the two.

Limitations.

There are two kinds of limitations of Galaxy-SynBioCAD.

First, Galaxy-SynbioCAD relies on the Galaxy software platform and then inherits from Galaxy limitations, which are:

- Available tools are limited to the ones registered in the main Tool Shed of Galaxy, and the process to publish a tool is a cumbersome (see key learning points for further details).
- It is strongly recommended to release additional tools as a Conda package. If not, additional work needs to be done in collaboration with developers to build such a package.
- The server side of client-server tools need to be hosted outside of Galaxy.
- Tools need to be executable from a command line interface, without the need for a graphical user interface.
- Tools have to be compliant with other tools in use, *i.e.* the use of standard file formats are of main importance.

The second kind of limitation is linked to the need for a minimum of knowledge in Synthetic Biology and Metabolic Engineering. The Galaxy-SynBioCAD portal offers access to users with a polished graphical interface, and preset workflows for general use cases are made available. Still, prior knowledge in the application fields are required from users in order to understand the concept and the parameters that matter in the processes (*e.g.* notion of “steps” for retrosynthesis, cell compartment, biomass reaction ID...). Considering this, and this is related to question 11 of reviewer 2, we believe a particular attention should be provided by developers to “surround” their developments with documentation, presentation and tutorials.

Key learning points.

For each tool the following should be considered:

- The tool has to be published in the main Tool Shed of Galaxy, and the process to publish a tool is a bit cumbersome (see added text in ACTION section of this question),
- An existing Conda package of the tool is recommended. If not, work needs to be done in collaboration with developers to build such a package.
- Following good programming guidelines when developing tools are of prime importance for its permanence and its integration in higher level systems like Galaxy. This is in particular true for organizing code, writing unit tests and documenting Input/Output.
- Making use of continuous integration services (*e.g.* GitHub Actions) enabling automation and systematization of tests after code change is a big time saver and a key for long term maintenance and keeping track of modifications that raise code incompatibilities.
- I/O standardization for interoperability. In our current Galaxy-SynBioCAD release some converters have been created and some software needed to be modified in collaboration with the developers.
- Clear and accessible documentation, step-by-step tutorials and even video presentations should not be neglected to help users become accustomed with the tools.

To guide our development process, we have followed a work process for each tool integrated within Galaxy-SynBioCAD. This work process is explained in the supplementary text (*Tools design and integration process*) and Figure S1.

ACTION

We added a new subsection in Supplementary Text (within *Tools design and integration process* section):

Publishing a tool in the Galaxy ToolShed

The publication of a tool is a three stages process as follow:

First, the source code is stored in a GitHub repository following a standard organization (see Figure S2). The 'README.md' file presents the purpose of the code, how to install and use it, and how to cite the software. The core source code is stored in a subfolder having the same name as the package. The 'setup.py' file, which includes all the package's metadata and structure, is used to specify how the package should be installed. 'MANIFEST.in' instructs setup tools on which files should be included when an installable project is created. Code test instructions are stored in a 'tests' directory, they are written using the pytest python library, which simplifies write, executions and reporting of unit tests. The 'LICENSE.txt' is added to specify the legal bindings for the use and the distribution of the code (typically an open source such as MIT, GPLv3...). The 'CHANGELOG.md' file lists in chronological order the history of changes.

Figure S2. Conda package file tree structure.

To help in the tool deployment and to simplify its installation, tools are packaged using the Anaconda environment management system¹⁶. Each package is built by adding a recipe file ('meta.yaml') in the 'recipe' subfolder which describes the package name, its version, its dependencies and how to build it (following conda-build guidelines²⁰).

Whenever a code change is committed, automatic tests are applied to the package through GitHub Actions and CircleCI (GitHub Continuous Integration Platform) which helps to automatically run build and test processes on Linux, MacOS, and Windows environments. The build status is reported on the GitHub branch. Packages are published on anaconda cloud in the conda-forge channel (whenever possible) or in the bioconda channel (as a fallback alternative) following instructions we provided on GitHub²¹ ('packaging_bioconda.md' and 'packaging_conda_forge.md' files).

Secondly, the Galaxy wrapper is created to describe the inputs, outputs and parameters of the packaged tool in XML format. This wrapper is tested using Planemo which is a command-line utility to assist in building and publishing Galaxy tools. For example, it helps to check the XML validity and to apply the tests that have been previously written in the wrapper's test section. Once the wrapper successfully passed the tests, the tool is published in (i) the Test Galaxy Tool Shed¹⁸, and then in (ii) the Main Galaxy Tool Shed¹⁹. The tool publication is done following instructions we provide on GitHub²¹ ('planemo_test_publish.md' file).

Finally, the Galaxy wrapper of each tool is submitted to the IUC (Intergalactic Utilities Commission) Galaxy community repository²². The submission passed through a critical review of Galaxy experts. The IUC also provides computational resources for automated testing through GitHub Actions and Tool Shed deployment²³.

In parallel to these steps, tutorials are released into Galaxy Training Platform²⁴ in a newly created category 'Synthetic Biology' for available Galaxy-SynBioCAD workflows. As an example, a published tutorial for the Genetic Design (Basic Assembly) workflow is available²⁵.

To support all of these steps and get help, it is worth mentioning Gitter²⁶, a chat and networking platform. We recommend using some specific Gitter chat rooms for (i) reaching the Galaxy community – Galaxy Training Network²⁷, Galaxy Project²⁸ – and (ii) reaching the Conda community – conda-forge²⁹, Bioconda³⁰.

10. As the tools selected are already published, what are the challenges faced in interfacing the different software? Was the current SBOL and SBML standard adequate to support the interfacing?

ANSWER

The main challenge in integrating the tools was that only few of them were released as a Conda package. To build a Conda package, either (i) developers were still working on the tool of interest and they were keen to make some modifications to their code or (ii) the tool development was stopped and we had to make the proper modifications.

SBOL is generally adequate to represent constructs and plasmids, however there are sometimes different ways of storing information. This is for instance the case for SBOL where SBOLv1, v2 and v3 formats coexist. For a given version of SBOL, there are also several ways of organizing the parts between them, for instance a genetic part could be defined as a subpart, ****or not****, of higher-level component definition. Hence this represents a difficulty for interoperability of tools.

The SBML format is used to represent strain models and pathways. As described in the section *Methods > Pathway annotation*, we have enriched the default format in order to store meaningful information for the pathways, especially annotations about detailed chemical structures of compounds and reactions, as well as reaction rule IDs and scores.

Additionally, some Galaxy-SynBioCAD tools do not process pathways and constructs and therefore do not read and write SBML or SBOL standard file formats but more generic ones (such as CSV, TSV, JSON, Python script).

11. In Discussion, can the authors provide suggestions on the key considerations for developers who are interested to contribute their software to the Toolshed?

ANSWER and ACTION

See question 9

REVIEWERS' COMMENTS

Reviewer #1 (Remarks to the Author):

Authors address my concerns well. The revised paper is now acceptable.

Reviewer #2 (Remarks to the Author):

The comments raised earlier have been addressed. I thank the authors for the clarifications.

I have one additional minor comment:

Regarding "A total of 88 construct designs were generated..", I would think it will be good to clarify how the 88 constructs were designed. Was it done manually or through using a software?

We thank the reviewers for their final comments. Below we address the one question that was raised by Reviewer #2 (under ANSWER) and also list the modifications we made to the manuscript (under ACTION).

Reviewer #1 (Remarks to the Author):

Authors address my concerns well. The revised paper is now acceptable.

Reviewer #2 (Remarks to the Author):

The comments raised earlier have been addressed. I thank the authors for the clarifications.

I have one additional minor comment:

Regarding “A total of 88 construct designs were generated..”, I would think it will be good to clarify how the 88 constructs were designed. Was it done manually or through using a software?

ANSWER

The generation of the 88 construct designs was not done manually but performed using the BasicDesign tool, which is part of the Genetic Design (BASIC) workflow. The construct have been generated in a fully automated fashion, from the following inputs: (i) the rpSBML file annotated with enzyme IDs for each reaction (ii) and several files listing the linker, promoter and backbone IDs.

The details of the operations performed by the Genetic Design workflow and the part IDs are provided in the “Genetic design and engineering workflow execution” section in Methods, while the details on how the BasicDesign tool works are provided in the supplementary text.

ACTION

We added the following in the main text “A total of 88 construct designs were automatically generated by the genetic design and engineering BasicDesign tool. The designs were coded in a .csv file that was fed to DNA-Bot [...]”.